# Validity evidence for assessing social-emotional psychological strengths in Colombian adolescents using the SEHS-S

Diana Riaño-Hernández[1], Iwin Leenen[2]*, Angelli Ramírez-Conde[3], Paula A. Atehortua-Rivera[3], José A. Piqueras[4]

1 Facultad de Psicología, Universidad del Valle, Valle del Cauca, Cali, Colombia, 2 Facultad de Psicología, Universidad Nacional Autónoma de México, Ciudad de México, México, 3 Facultad de Ciencias Humanas, Sociales y de la Educación, Universidad Católica de Pereira, Risaralda, Pereira, Colombia, 4 Departamento de Psicología de la Salud, Universidad Miguel Hernández, Elche, Alicante, España

* iwin.leenen@unam.mx

**Data Availability Statement:** The data from which all results presented in this study are derived are publicly available in different file formats from a

## Abstract

### Background

Covitality is a multidimensional hierarchical construct of core psychological strengths that synergistically promote resilience and well-being and that has been shown to be effective in preventing mental health problems in individuals of different age groups. The Covitality Model consists of 12 first-order latent factors, 4 second-order factors, and one general higher-order Covitality factor.

### Purpose

In this study, we aim at obtaining validity evidence for the assessment of Covitality in Colombian adolescents by means of the Social Emotional Health Survey-Secondary (SEHS-S).

### Method

A sample of 1461 adolescents responded the SEHS-S and four other instruments that measure well-being and distress. The internal structure of the SEHS-S was examined through confirmatory factor analyses as well as its relations with other variables.

### Results

The hierarchical factor structure of the SEHS-S was supported (with goodness-of-fit statistics: $\chi^2 = 1727.6$, $df = 578$, p < .001; RMSEA = .037; SRMSR = .044; AGFI = .962; CFI = .940; and NNFI = .935) and configural and metric invariance across gender and age was confirmed; however, the assumption of scalar invariance across males and females and across age groups was violated for some items. Furthermore, we found moderate to high correlations ($r = .56 –.68$) of Covitality with related constructs.

project registered at the Open Science Framework with doi: 10.17605/OSF.IO/ND7HU.

**Funding:** This research received funding from the Universidad Católica de Pereira (Project 2019/018) and the Colombian Ministerio de Ciencia y Tecnología (DRH; Project 2019/850). The funders had no role in study design, data collection and analysis, decision to publish, or preparation of the manuscript.

**Competing interests:** The authors have declared that no competing interests exist.

## Conclusion

As a conclusion, the SEHS-S can be considered a valid tool to assess psychological strengths, well-being, and resilience (i.e., Covitality) in Colombian adolescents, though further research is needed to explore the differences in item functioning across gender and age.

## Introduction

The 2015 National Mental Health Survey in Colombia highlighted that around 12% of the adolescents showed signs of emotional or mental health symptoms [1] and 2.8% presented high-risk alcohol abuse [2]. Moreover, suicide rates among children and adolescents increased substantially, with minors representing 10.5% of the overall national figure [3]. Yet, Colombia's public health system lacks the resources to provide adequate psychological support, preventative initiatives have limited reach, and private psychological services are unaffordable for most of the population [4].

Developing strategies that identify and address emotional and behavioral issues early, with a focus on proactive prevention, is crucial [5, 6]. Many authors have studied mental health in children and adolescents and, as such, provided evidence supporting the Bidimensional Model of Mental Health [7–10]. This model, rather than viewing psychological well-being as the absence of mental dysfunction, considers well-being and psychopathology as separate (but connected) dimensions that work together to establish and maintain mental health [11–14]. This insight led to a fundamental shift, not only in the understanding but also in the treatment of mental health problems [15–18]. Interventions should contribute to socio-emotional well-being by fortifying the individual's strengths and protective factors directly, besides minimizing psychosocial risk factors and symptoms of psychological damage [19–21]. As an example, a study by Martinotti and colleagues offers a comprehensive analysis of alcohol consumption patterns among young people and shows that gaining a deeper understanding of these behavioral patterns and psychological strengths of adolescents is beneficial for developing effective interventions, such as those aimed at preventing binge drinking [22].

In this context emerged the meta-construct of covitality — the theoretical counterpart of comorbidity —, which is defined as "the synergistic effect of positive mental health resulting from the interplay among multiple positive-psychological building blocks (. . .) [or] more technically as the latent, second-order positive mental health construct accounting for the presence of several co-occurring, first-order positive mental health indicators" [23]. That is, the covitality model does not just rely on individual psychological abilities but rather on developing and combining a large number of strengths and actives, with special emphasis on their interplay and conjunction. These psychological skills are then further consolidated into cognitive schemes that organize and give meaning to life experiences and that gain importance as they foster positive development and turn into protective factors that help to overcome emotional distress [21, 23, 24]. Covitality must be understood as the joint effect and interaction of these cognitive schemes, which lead directly to subjective well-being [25, 26].

Prevention strategies rooted in socio-emotional strengths emphasize the importance of fostering positive mental health through early, school-based interventions [27–29]. Educational institutions provide a natural environment for promoting psychological well-being and programs that cultivate socio-emotional skills are crucial for both prevention and promotion of mental health. Examples include Positive Psychology Interventions, which aim to build skills

such as self-regulation, coping, and interpersonal communication [30]. These can be implemented as part of broader school culture programs or targeted individual support, fostering resilience and emotional well-being among students. Additionally, transitional support programs, such as pre- and post-transfer interventions during school transitions, help students adjust to new educational environments by reinforcing a positive mindset and emotional preparedness. Programs like the "Student Strengths Safari" [31] or Growth Psychoeducation Interventions (GPI) [32] focus on developing covitality by enhancing executive functioning and social-emotional skills. These efforts underline the importance of early intervention and a supportive school community in fostering socio-emotional strengths.

Furlong and colleagues developed a self-report measure called the Social Emotional Health Survey (SEHS-S) to assess covitality in adolescents. This measure is based on a $1 \rightarrow 4 \rightarrow 12 \rightarrow 36$ model of 36 items organized in 12 subscales that load onto four domains, culminating into one higher-order covitality latent construct (see the Instruments section below for further details). The SEHS-S has gained worldwide interest and validity evidence has been obtained for its use in various cultures and contexts. The original studies by Furlong and colleagues [33–36] with Californian middle and high school students validated the SEHS-S internal structure using confirmatory factor analyses (CFA), which subsequently was confirmed by studies in Japan, Korea, China, Lithuania, Turkey, Iran, and Spain [5, 6, 18, 37–41]. Further studies added evidence for measurement invariance across gender, age, and ethnic groups [42, 43] and others showed positive associations of the higher-order covitality construct with subjective well-being, resilience, prosocial behavior, quality of life, and school adjustment, and mental health and negative associations with psychopathology [5, 6, 18, 37, 44, 45]. It is worth mentioning that in 2020, Furlong et al. proposed an updated version of the SEHS-S, with a standardized four-point response scale for all 36 items and minimal changes to enhance readability.

In this paper, we provide validity evidence for using the SEHS-S with Colombian adolescents by studying its internal structure (factorial structure, measurement invariance and reliability) and relations with external variables. Currently in Colombia mental health is still approached from the traditional model, with a focus on negative indicators such as anxiety, and instruments based on individual's strengths and protective factors are rarely used or even unavailable. In that sense, the SEHS-S can be used to shed a light on the assets and socio-emotional abilities of Colombian adolescents and how they contribute to maintaining their mental health; the needs of these students for an adequate socio-emotional development may be identified more precisely and more efficient intervention strategies may be proposed to increase their well-being and prevent or reduce psychosocial and behavioral problems. As such, this study contributes to the body of validity evidence for the use of the SEHS-S in different countries and cultures.

## Materials and methods

### Participants

A total of 1,473 students belonging to four public and rural secondary schools from the Colombian department of Risaralda volunteered to participate in this study. These schools were selected by convenience, for their proximity, accessibility, and for being the largest schools in the department. To be eligible for participation, students had to demonstrate basic reading and writing skills and be free from any diagnosed mental or cognitive disabilities. These inclusion and exclusion criteria were verified by both the students' teachers and the psychologists at the participating schools.

Of the 1,473 students who participated, 12 were excluded due to largely incomplete responses on the instruments (i.e., over 50% of missing responses overall). Note that, following recommendations and common practice regarding sample size in factor analysis [46, 47], the remaining 1,461 participants are sufficient for the planned data analysis.

The students were about evenly distributed between sixth and eleventh grade, all were between 10 and 19 years old (mean: 14.3 years; with 95% between 11 and 17 years), and 44% were women. The participants largely resided in small municipalities with low socioeconomic status and their primary livelihoods revolving around activities such as farming, fishing, and agriculture.

## Instruments

**Social emotional health survey–Secondary.** The SEHS-S is a self-report questionnaire for adolescents of between 12 and 18 years old. The 36 items are organized in 12 subscales (with each subscale consisting of three items and measuring one positive psychological component), which in turn are grouped in four domains of positive mental health: belief-in-self (with the subscales self-efficacy, self-awareness, and persistence), belief-in-others (school support, family coherence, peer support), emotional competence (emotional regulation, empathy, behavioral self-control), and engaged living (gratitude, zest, optimism). These four domains contribute to the overall construct of covitality. Items are responded using a four-point Likert-type format (1 = "not at all true of me", 2 = "a little true of me", 3 = "more or less true of me", and 4 = "very much true of me"). We used Piqueras's [18] translation into Spanish of the SEHS-S, with the adjustments by Furlong [33].

**Kidscreen–10 Index.** The Kidscreen–10 Index [48] assesses the subjective quality of life — related to health and well-being during the week preceding the application — of children and adolescents between 8 and 17 years. It consists of 10 items, with items 1 and 2 evaluating the child/adolescent's level of physical activity, condition and energy, items 3 and 4 their current mood and unpleasant emotions, items 5 and 6 their freedom of choice about entertainment and social activities, items 7 and 8 their relationship with parents (or caregivers) and peers, and items 9 and 10 their perception of the own cognitive abilities and academic achievement. Responses are given using a five-point Likert-type format (1 = "never", 2 = "almost never", 3 = "sometimes", 4 = "almost always", 5 = "always"). According to a recent review, this instrument has been used up to six times previously in the Colombian population with adequate psychometric properties [49].

**The MHI-5 mental health inventory.** The MHI-5 [50] consists of five (two positive and three negative) items asking the respondent about how they felt during the last month. The instrument has been translated to multiple languages and can rely on validity studies in many countries, including Brazil [51], Mexico [52], Spain [53], and Peru [54]. Responses are given using a four-level Likert-type format (with 0 = "never", 1 = "sometimes", 2 = "many times", 3 = "always").

**The Trait Emotional Intelligence Questionnaire–Adolescent Short Form (TEIQue–ASF).** The Spanish version of the TEIQue-ASF [55], which was retrieved from http://www.psychometriclab.com, is a simplified version of the abbreviated form of the TEIQue global trait emotional intelligence measure for adults and comprises 30 short statements (two for each of the 15 facets of emotional intelligence) with a seven-point Likert-response format (1 = "never" to 7 = "always"). Ferrando et al. [56] obtained adequate psychometric properties for the instrument in Spanish adolescents.

**Pediatric Symptom Checklist-Youth self-report (PSC-17-Y).** The short form of the PSC-17-Y [57] assesses psychosocial problems, overall and in three main psychopathological

domains: internalizing (anxiety and depression), externalizing (disruptive behavior) and attention deficit hyperactivity disorder (ADHD). It consists of 17 items, scored on a three-point Likert scale (0 = "never", 1 = "sometimes", 2 = "always"). Piqueras et al. [58] provided validity evidence for the instrument in Spanish adolescents.

## Procedure

Cultural relevance, clarity, comprehensibility, and potential ambiguities in the SEHS-S questionnaire by Piqueras [18] were evaluated by three expert judges and through cognitive interviews with five adolescents from the target population; based on their suggestions, slight adjustments to 22 of the 36 items were made. After obtaining informed consent (see the "Ethical approval and informed consent" section below), the instruments and a sociodemographic questionnaire were administered in groups. The data collection took place on selected dates between March 1 and October 17, 2019. Different collaborators visited all four schools on each of the selected dates and administered the instruments to all students from a given grade who were present at school that day and had provided their consent.

## Data analysis

We calculated descriptive statistics (means, standard deviations, intercorrelations, Cronbach's alpha, and McDonald's omega) for the scale scores (i.e., sums across all items) of the five instruments. Additionally, for the SEHS-S, we present descriptive statistics (including percentiles) separately for gender (male versus female) and age group (grades 7–8 versus grades 9–10 versus grades 11–12). Given the relatively large number of individuals with at least one missing value, we employed multiple imputation (with 25 imputed data sets) for the missing values based on the fully conditional specification (FCS) method and multivariate regression on the full set of response variables, assuming data are missing at random (MAR) [59]. We used the PROC MI and PROC MIANALYZE procedures of SAS V9.4 [60].

Subsequently, we examined the internal structure of the SEHS-S adopting an approach similar to the one used by previous validation studies (see the Introduction). We ran CFA on the 36 response variables to fit the three-level higher-order factorial structure of the SEHS-S. To evaluate goodness of fit, we report the chi-square statistic (with associated degrees of freedom), the root mean square error of approximation (RMSEA), the standardized root mean square residual (SRMSR), the adjusted goodness-of-fit index (AGFI), Bentler's comparative fit index (CFI), and the Bentler-Bonett nonnormed fit index (NNFI). With the chi-square test being highly sensitive to sample size, we consider values below 0.05 for RMSEA and SRMSR and values above 0.95 for AGFI, CFI, and NNFI excellent fit, while values between 0.05 and 0.08 for RMSEA and SRMSR and between 0.90 and 0.95 for CFI and NNFI are still acceptable [61].

The invariance assumptions for the obtained factor model were tested across gender and age groups (as defined above), considering three levels of invariance: configural invariance (which is confirmed if the above model fits well in each subpopulation), metric invariance (which holds if, additional to configural invariance, the factor loadings are equal across subpopulations), and scalar invariance (which means that, additional to metric invariance, variable intercepts are equal across subpopulations). Following Cheung and Rensvold [62], metric and scalar invariance were tested by comparing a restricted model (equal loadings and/or intercepts) with the more general model (unrestricted loadings and/or intercepts) through the CFI of both models, with a difference (ΔCFI) of .01 or less indicating that the null hypothesis of invariance can be maintained. All factor models were fitted by the SAS V9.4 PROC CALIS procedure [60] specifying full-information maximum likelihood estimation and Levenberg-

Marquardt optimization. Note that this software is known for its robustness and ability to handle large datasets with missing values.

## Ethical approval and informed consent

This study was reviewed and formally approved by the Ethics Committee of the Universidad Católica de Pereira, Colombia, on November 19, 2018. The research procedures were conducted in strict accordance with the guidelines and principles outlined in the Declaration of Helsinki. Participant schools sent detailed information about the project to parents, asking permission for their adolescent children to participate in the study. Both parents and adolescents provided written informed consent before completing the questionnaires. Furthermore, all data were analyzed anonymously and the study was designed to ensure the safety and well-being of all subjects involved.

## Inclusivity in global research

Additional information regarding the ethical, cultural, and scientific considerations specific to inclusivity in global research is included in the Supporting Information (S1 Checklist).

## Results

### Descriptive statistics

Table 1 shows means, standard deviations, and reliability statistics through Cronbach's alpha and McDonald's omega (with their respective standard errors) for the sum scores on each of the five instruments applied to the 1,461 participants. Given the higher-order factor structure for the SEHS-S, we also calculated McDonald's hierarchical omega [63]: $\omega_H = .923$ (with a standard error of .025). Furthermore, the table reports the intercorrelations (with 95%-confidence intervals) among scales and the percentage of nonresponse in each scale. Whereas the overall nonresponse rate is low (considering all scales, 1.0% of the item responses were missing), the percentage of individuals who did not respond to at least one item is relatively high, especially in the longer instruments (viz., 22.7% for the TEIQue-ASF, the highest among the instruments used). As explained in the Data Analysis section, missing responses were imputed under the missing-at-random assumption. Given this assumption and the relatively low nonresponse rate at the item level, their impact on the results and conclusions is expected to be minimal. For the SEHS-S, Table 2 presents percentile scores, separated by gender and age groups.

### Factorial structure and invariance

Standardized loadings for Furlong et al.'s [33] three-order factor model, estimated by a CFA on the sample SEHS-S data, are shown in Fig 1. The model had a good to excellent fit to the data, with $\chi^2 = 1727.6$ ($df = 578$, p < .001), RMSEA = .037, SRMSR = .044, AGFI = .962, CFI = .940, and NNFI = .935.

Next, we tested the model for invariance. First, we checked invariance across gender. Configural invariance was confirmed by separately fitting the model to males and females and obtaining adequate fit indices in both subsamples (males: $\chi^2 = 1217.8$, RMSEA = .037, SRMSR = .047, AGFI = .953, CFI = .933, NNFI = .927; females: $\chi^2 = 1134.3$, RMSEA = .038, SRMSR = .046, AGFI = .949, CFI = .942, NNFI = .937). As to metric invariance, a model with the factor loadings being restricted to be equal for males and females did not fit substantially worse than a more general model that allowed for different factor loadings in both groups ($\Delta$CFI = .001; see Table 3 for details); consequently, we maintained the null hypothesis of metric invariance.

**Table 1. Means, standard deviations, percentage of nonresponse, intercorrelations, Cronbach's alpha and McDonald's omega for the five scales applied to the sample of 1461 adolescents.**

|  | SEHS-S (36 items) | Kidscreen-10 (10 items) | MHI-5 (5 items) | TEIQue-ASF (30 items) | PSC-17-Y (17 items) |
|---|---|---|---|---|---|
| *Mean* | 116.03 ± 0.41 | 38.63 ± 0.15 | 9.91 ± 0.07 | 139.78 ± 0.62 | 11.64 ± 0.14 |
| *Standard deviation* | 15.59 ± 0.29 | 5.74 ± 0.11 | 2.85 ± 0.05 | 23.70 ± 0.44 | 5.51 ± 0.10 |
| *Nonresponse* |  |  |  |  |  |
| Overall[a] | 0.6% | 0.7% | 0.5% | 1.7% | 0.9% |
| Cases[b] | 14.1% | 4.8% | 1.4% | 22.7% | 7.3% |
| *Intercorrelations* |  |  |  |  |  |
| SEHS-S |  | .679 [.651,.706] | .556 [.520,.591] | .662 [.632,.690] | −.563 [−.598,−.527] |
| Kidscreen-10 |  |  | .714 [.688,.739] | .618 [.585,.649] | −.552 [−.587,−.515] |
| MHI-5 |  |  |  | .602 [.568,.633] | −.551 [−.586,−.514] |
| TEIQue-ASF |  |  |  |  | −.609 [−.640,−.576] |
| *Cronbach's alpha* | .911 ± .003 | .782 ± .008 | .786 ± .009 | .834 ± .006 | .800 ± .008 |
| *McDonald's omega* | .908 ± .005 | .787 ± .009 | .789 ± .009 | .824 ± .008 | .803 ± .008 |

*Notes.* For the mean, standard deviation, Cronbach's alpha and McDonald's omega, standard errors are reported, while the uncertainty about the intercorrelations is accounted for by 95%-confidence intervals (based on a Fisher-$Z$ transformation of the correlations). For the standard errors for McDonald's omega, bootstrapping was used [64]. All statistics are based on 25 multiply-imputed data sets.

[a]Overall nonresponse is the percentage of cells with missing values in the data matrix

[b]Cases nonresponse is the percentage of individuals who did not respond one or more items of the scale.

Finally, with respect to scalar invariance, we did find a better fit for the model that allowed for different intercepts across genders as compared to a model that restricts these intercepts to be equal (ΔCFI = .018). To identify the items with significantly different intercepts for males and

**Table 2. Percentile scores for the SEHS-S, overall and by gender and age groups.**

|  |  | Grades 6–7 |  | Grades 8–9 |  | Grades 10–11 |  |
|---|---|---|---|---|---|---|---|
| **Percentiles** | **Overall** | **Females** | **Males** | **Females** | **Males** | **Females** | **Males** |
|  | (*n* = 1461) | (*n* = 282) | (*n* = 332) | (*n* = 152) | (*n* = 206) | (*n* = 214) | (*n* = 275) |
| 1 | 72 | 62 | 76 | 69 | 74 | 73 | 76 |
| 5 | 88 | 87 | 92 | 84 | 90 | 86 | 89 |
| 10 | 95 | 92 | 100 | 93 | 94 | 92 | 97 |
| 15 | 100 | 100 | 103 | 96 | 99 | 96 | 101 |
| 20 | 104 | 105 | 106 | 101 | 104 | 100 | 103 |
| 25 | 107 | 109 | 109 | 103 | 105 | 105 | 107 |
| 30 | 110 | 112 | 110 | 106 | 108 | 109 | 110 |
| 40 | 114 | 117 | 115 | 111 | 113 | 111 | 115 |
| 50 | 117 | 121 | 118 | 114 | 117 | 115 | 117 |
| 60 | 121 | 126 | 123 | 118 | 121 | 120 | 120 |
| 70 | 126 | 130 | 127 | 122 | 125 | 124 | 123 |
| 75 | 127 | 132 | 129 | 125 | 126 | 125 | 125 |
| 80 | 130 | 134 | 131 | 127 | 128 | 127 | 127 |
| 85 | 132 | 136 | 134 | 131 | 131 | 130 | 129 |
| 90 | 135 | 138 | 136 | 132 | 133 | 132 | 132 |
| 95 | 138 | 140 | 139 | 137 | 136 | 135 | 136 |
| 99 | 143 | 143 | 144 | 144 | 141 | 139 | 140 |

*Note.* All percentiles are based on 25 multiply-imputed data sets.

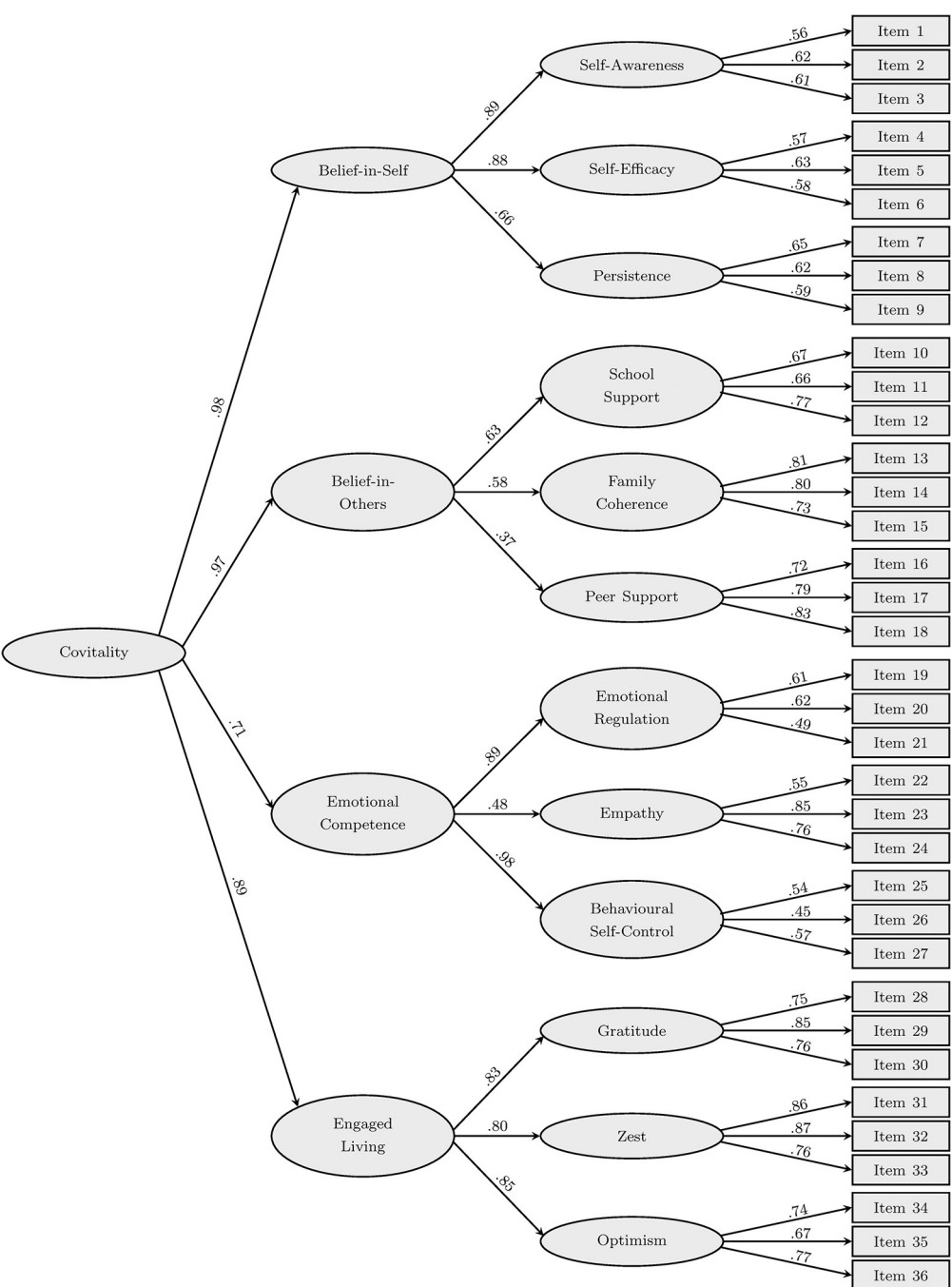

**Fig 1. Standardized factor loadings from a confirmatory factor analysis on the SEHS-S (Furlong et al.'s [33] three-order factor model).**

females, we tested the intercept parameters one-by-one in a stepwise procedure to obtain a stable division of the items in two groups: one with intercepts that were significantly different for males and females and the other where this is not the case. Table 4 shows the results of this procedure. Refitting the model with these partially constrained intercepts yielded an almost equal fit as compared to the model with unconstrained intercepts (ΔCFI = .001, between Model G2 and Model G4 in Table 3).

**Table 3. Goodness-of-fit indices for models used to test invariance across gender and age groups.**

|  | $\chi^2$ (df) | RMSEA | SRMSR | AGFI | CFI | NNFI |
|---|---|---|---|---|---|---|
| *Gender* | | | | | | |
| Model G1 | 2352.2 (1158) | .038 | .047 | .951 | .937 | .932 |
| Model G2 | 2410.8 (1192) | .037 | .051 | .951 | .936 | .933 |
| Model G3 | 2786.8 (1227) | .042 | .056 | .949 | .918 | .916 |
| Model G4 | 2439.9 (1206) | .038 | .051 | .951 | .935 | .933 |
| *Age groups* | | | | | | |
| Model A1 | 3202.9 (1737) | .042 | .055 | .936 | .925 | .918 |
| Model A2 | 3309.7 (1807) | .041 | .062 | .935 | .923 | .919 |
| Model A3 | 3645.4 (1878) | .044 | .065 | .935 | .909 | .909 |
| Model A4 | 3398.7 (1851) | .041 | .063 | .936 | .921 | .919 |

*Notes.* Model G1/A1: Model with unequal factor loadings and unequal intercepts across gender/age groups; Model G2/A2: Model with equal factor loadings and unequal intercepts across gender/age groups; Model G3/A3: Model with equal factor loadings and equal intercepts across gender/age groups. Model G4/A4: Model with equal factor loadings and partially unequal intercepts (see, Table 4) across gender/age groups. In all models, unique variances are allowed to differ across gender/age groups. Gender refers to two groups (males and females); age groups to three groups (Grades 6–7, Grades 8–9, Grades 10–11).

The SEHS-S items are scored on a scale from 1 to 4 points; therefore, the intercepts indicate the expected score for an individual with average latent factor scores. In Table 3, we read for instance that on item 17 ("I have a friend of my age who talks with me about my problems"), a female is expected to have a score that is almost 0.5 points higher than a male who has identical scores on the Covitality constructs and subconstructs; in other words, a high score on that item is not as indicative of high covitality in females as it is in males. Likewise, in item 5 ("I understand my moods and feelings") a male is expected to have a score of almost 0.3 points higher than a female with the same level of self-efficacy, belief-in-self, and covitality; that is, a high score on this item is more indicative of high covitality in females that in males. Interestingly, when considering all items in the scale and summing all items into the overall test score, it turns out that the expected difference between males and females with the same scores on the latent factors largely cancels out and is almost negligible as females are expected to score only 0.74 points (on a scale that ranges from 36 to 144) higher as compared to males of the same covitality level. This means that, for a fair comparison of males and females (e.g., in Table 2) test scores for females should be lowered (or scores for males increased) with 0.74 points.

We further checked invariance across age groups. Again, configural invariance (with adequate fit in the three age groups; grades 6–7: $\chi^2$ = 1054.6, RMSEA = .036, SRMSR = .046, AGFI = .945, CFI = .938, NNFI = .932; grades 8–9: $\chi^2$ = 1078.1, RMSEA = .049, SRMSR = .059, AGFI = .912, CFI = .899, NNFI = .890; grades 10–11: $\chi^2$ = 1,070.2, RMSEA = .042, SRMSR = .061, AGFI = .943, CFI = .929, NNFI = .923) as well as metric invariance hold (with ΔCFI = .002 between a model where factor loadings are restricted to be equal across age groups versus a model where this is not the case, see Table 3), whereas the assumption of scalar invariance is violated: A model with unconstrained intercepts fits substantially better than a model with the intercepts being constrained to be equal across age groups (ΔCFI = .014). Identifying the subset of items for which a constrained intercept leads to a significantly worse fit (using a similar stepwise procedure as the one described above), yields the results shown in the last columns of Table 4. The model with partially restricted intercepts does not fit substantially worse than the model with free intercepts (ΔCFI = .002, between Model A2 and Model A4 in Table 3). The interpretation follows the same lines as in the case of gender non-invariance for the intercepts.

**Table 4. Estimated intercepts for the 36 items in the higher-order factor model for the SEHS-S, by gender and age groups.**

| | Gender | | Age groups (grades) | | |
|---|---|---|---|---|---|
| Item | Female | Male | 6 – 7 | 8–9 | 10–11 |
| 1 | 3.22 | 3.32 | 3.28 | 3.36 | 3.54 |
| 2 | — 3.46 — | | 3.50 | 3.58 | 3.69 |
| 3 | — 3.24 — | | 3.29 | 3.31 | 3.49 |
| 4 | 3.68 | 3.59 | 3.67 | 3.74 | 3.80 |
| 5 | 2.85 | 3.12 | — 3.16 — | | |
| 6 | 3.08 | 3.20 | 3.18 | 3.30 | 3.38 |
| 7 | — 2.59 — | | — 2.72 — | | |
| 8 | 2.81 | 2.90 | 3.10 | 2.99 | 2.81 |
| 9 | 2.70 | 2.80 | 2.98 | 2.79 | 2.79 |
| 10 | — 3.35 — | | — 3.46 — | | |
| 11 | — 3.00 — | | — 3.13 — | | |
| 12 | — 3.23 — | | — 3.36 — | | |
| 13 | — 3.43 — | | — 3.55 — | | |
| 14 | 3.30 | 3.35 | — 3.45 — | | |
| 15 | — 3.33 — | | — 3.44 — | | |
| 16 | 3.29 | 2.97 | — 3.20 — | | |
| 17 | 3.22 | 2.74 | 2.91 | 3.12 | 3.17 |
| 18 | 3.27 | 2.91 | — 3.17 — | | |
| 19 | — 3.45 — | | 3.45 | 3.48 | 3.67 |
| 20 | — 3.31 — | | 3.31 | 3.36 | 3.56 |
| 21 | — 3.08 — | | 3.10 | 3.13 | 3.26 |
| 22 | 3.34 | 3.06 | — 3.23 — | | |
| 23 | 3.44 | 3.19 | — 3.37 — | | |
| 24 | 3.36 | 3.11 | — 3.29 — | | |
| 25 | — 3.26 — | | — 3.36 — | | |
| 26 | 3.00 | 2.89 | — 3.03 — | | |
| 27 | 2.73 | 2.93 | — 2.95 — | | |
| 28 | 3.25 | 3.41 | — 3.48 — | | |
| 29 | — 3.42 — | | 3.49 | 3.61 | 3.64 |
| 30 | — 3.49 — | | — 3.61 — | | |
| 31 | 2.95 | 3.18 | — 3.24 — | | |
| 32 | 3.00 | 3.17 | — 3.26 — | | |
| 33 | 2.95 | 3.09 | — 3.17 — | | |
| 34 | — 3.39 — | | — 3.52 — | | |
| 35 | 3.38 | 3.22 | — 3.42 — | | |
| 36 | 3.65 | 3.58 | 3.68 | 3.73 | 3.77 |
| **Sum score** | **115.52** | **114.78** | **118.52** | **119.08** | **120.13** |

*Note*. If only one intercept is reported for both genders or for the three age groups, the intercepts were constrained to be equal after a previous hypothesis test showed that these intercepts were not significantly different (i.e., $p > .05$). In this model, factor loadings were restricted to be equal across groups.

Item 8 ("I try to respond all questions that are asked us during the lessons") shows the largest difference between the groups: Adolescents of grades 6–7 are expected to score about .30 points higher than adolescents of grades 10–11 who have the same scores on the latent Covitality construct and subconstructs. Conversely, for item 17, older adolescents are expected to score about .26 points higher as compared to younger adolescents of the same covitality level:

Having a friend with whom you can talk about your problems is less indicative of high covitality in older adolescents. Summing all the intercepts shows that the overall test score is only slightly biased, with adolescents in grades 10–11 having an advantage of 1.61 points over adolescents from grades 6–7 and an advantage of 1.05 points over adolescents of grades 8–9.

## Gender and age differences

As explained in the previous section, differences in Covitality between males and females or among adolescents of different age can be examined through the overall test score, after taking into account the above mentioned differences due to lack of scalar invariance. A more direct comparison (using the CFA results) is based on the estimated latent scores on the Covitality super-factor. With respect to sex, we found that, on average, men have a slightly higher latent score than women (of 0.044, on a scale with a standard deviation of 0.39 for women and 0.33 for men; $p = .047$). As to age groups, the average latent Covitality score for adolescents in grades 6–7 is 0.130 units higher than for those in grades 8–9 and 0.149 higher as compared to adolescents of grades 10–11 ($p < .01$ for both differences; the standard deviations for the three groups are 0.38, 0.37, and 0.33, respectively).

## Discussion

Covitality refers to the positive aspects of mental health, such as well-being, positive emotions, and social support [20, 22–24]. In this study, we provide a validation of the SEHS-S, the most wide-spread measure of covitality, in a population of Colombian adolescents aged 10 to 19 years. The results provide validity evidence based on the SEHS-S internal structure (including factor structure, measurement invariance, and reliability) as well as on its relations with external variables.

Regarding the descriptive statistics, we found that the mean in our sample was slightly above the mean reported in Furlong et al.'s [34] recent study. However, further research is required to interpret this difference because, in the first place, cross-cultural measurement invariance should be established (e.g., equivalence with respect to the instrument used in both populations) and then, possibly, this difference may be validly attributed to differences in the living conditions between Californian and Colombian adolescents. Furthermore, Furlong et al. [33] consider SEHS-S scores between percentiles 16 and 84 as indicative of normal covitality, whereas scores below the 15th percentile and above the 85th percentile are regarded as "weak" and "strong" covitality, respectively. Translated to the population in this study, these percentiles correspond to overall sum scores of 100 and 132, respectively, which may be relevant cut-offs when the SEHS-S is used for educational or clinical purposes in Colombia.

The CFAs in this study support the a priori hierarchical factor structure of the super-construct Covitality, where Covitality is as a third-order factor, composed of 4 second-order factors and 12 first-order factors measured by 36 items. As such, our results are in line with the original Covitality model, which has been corroborated in a wide variety of other cultures and contexts [18, 33, 34, 41, 43]. This suggests that configural invariance of the Covitality model across cultures is plausible.

Furthermore, this study examined measurement invariance across different subpopulations, specifically across boys and girls and across age groups (based on the participants' school grade). We found that, whereas the hierarchical factor structure fits well in both male and female subpopulations as well as in three different age groups, and both configural and metric invariance was confirmed, differences in the intercepts for different items pointed to a lack of scalar invariance. This lack of scalar invariance means that two individuals from different genders and/or from different age groups, although they have equal scores on the latent covitality

factors, have different expected scores for at least some of the items. Examples include item 17 ("I have a friend my age who talks to me about my problems", which is more common in girls and in older adolescents), item 5 ("I understand my moods and feelings", more common in boys) and item 8 ("I try to answer all the questions asked in class", more common in younger students). It is important to take this lack of invariance into account as it implies that the same response is more indicative of covitality in some subpopulations than in others (e.g., item 17 points to a higher level of covitality if endorsed by a younger boy as compared to an older girl). In this respect, our results are notably different from most previous work with the SEHS-S, where scalar invariance as a function of gender and age received support [18, 33, 34, 43]. Overall, our findings suggest that while the SEHS-S can be validly used to measure social and emotional health in Colombian adolescents, there are (small) differences in response patterns between boys and girls and adolescents of different age that are worthwhile to be further investigated. In general, it is important to test for measurement invariance in different subpopulations when using self-report measures of social and emotional health and to be aware of potential biases in the interpretation of results.

With respect to the reliability of the SEHS-S, the values for Cronbach's alpha and McDonald's omega we obtained in this study were high (above .90), which means that the SEHS-S is a reliable measure in this population of Colombian adolescents. Also within subgroups defined by sex and age the values for Cronbach's alpha were high (between .89 and .93). These results are fully in line with previous studies, which report reliability indices in the range of .89 – .96 [5, 6, 18, 23, 33–36, 38].

Further validity evidence is provided by the moderate to strong associations of the SEHS-S with other variables that measure theoretically related constructs. In particular, our results show that covitality has positive and strong correlations with health-related quality of life and trait emotional intelligence, moderate positive correlations with well-being, and moderate negative correlations with psychopathological symptoms. These correlations are consistent with results from previous studies which report associations with similar measures of mental health and well-being [5, 6, 18, 33, 37, 41, 44, 45, 65].

Finally, our study explored differences on the overall Covitality construct both between males and females and among individuals of different age. Regarding gender, we found slightly higher social and emotional skills in boys, a result that is consistent with previous studies; in effect, whenever gender differences come across, it turns out that males are more likely to have high scores on global Covitality [5, 18, 23, 38]. Regarding age, our results reveal a trend of lower Covitality scores in older individuals. Although we have not found any other studies that have compared the level of global Covitality as a function of age or grade, this result is in line with the general decrease —reported for both boys and girls, although more pronounced in girls— of subjective well-being in adolescents with increasing age, starting at 11–12 years [66]. Some authors attribute this decline in well-being to the onset of adolescence, which is characterized by significant physical, cognitive, emotional, and social changes, as well as by an increase in risky situations that lead to greater emotional, psychological, and social vulnerability [67]. These gender and age differences are relevant for the development and implementation of interventions aimed at improving social and emotional health, both in general and in this population of Colombian adolescents. However, further research, especially with a longitudinal design, is necessary to examine how covitality evolves over time, potentially differing between boys and girls.

## Limitations

Our study faces the following limitations: In the first place, we used a convenience sample of adolescents from four specific schools in a particular area in Colombia, characterized by its

rather low social economic status. Moreover, data collection took place in 2019, before the COVID-19 outbreak. Therefore, any generalizations of the obtained results, whether in space (to other regions inside or outside Colombia) or time (e.g., post-COVID), are highly speculative.

Furthermore, the self-report nature of the instruments poses a potential risk of bias, as adolescents may struggle to accurately assess their own emotional states or may be reluctant to share their feelings. Studies that correlate adolescents' responses on the SEHS-S with qualitative data from in-depth interviews may shed a light on the extent to which participants' self-report measures on this instrument overestimate their actual social-emotional strengths. Conversely, Corneille and Gawronski, in a recent paper, highlight several advantages of self-report measures and argue for their superiority over implicit measures [68].

Finally, our analytic approach is fully quantitative and focuses exclusively on validity evidence based on internal structure and relations with other variables. Future research may include other sources of validity evidence, such as those based on response processes and consequences, and employ qualitative techniques, such as cognitive interviews or focus groups [69].

## Conclusions

Overall, the findings of this study support the use of the SEHS-S as a tool for promoting positive mental health and assessing social and emotional skills in adolescents. The hierarchical structure of the Covitality super-construct provides a comprehensive framework for understanding these skills. Although future research should examine gender and age differences, the results from the CFAs, high reliability indices, and correlations with similar constructs provide ample evidence for the SEHS-S as a valid tool for assessing social and emotional skills in Colombian adolescents. This can help develop targeted interventions to promote Covitality in this population.

## Supporting information

**S1 Checklist. Inclusivity in global research.**
(DOCX)

## Acknowledgments

We gratefully acknowledge the invaluable support received from the educational institutions Héctor Ángel Arcila, Instituto Santuario INSA, Nuestra Señora de la Presentación, and Institución Educativa Francisco José de Caldas, as well as from the Secretaria de Educación of the department of Risaralda, in facilitating the data collection process.

## Author Contributions

**Conceptualization:** Diana Riaño-Hernández, José A. Piqueras.

**Data curation:** Iwin Leenen.

**Formal analysis:** Iwin Leenen.

**Funding acquisition:** Diana Riaño-Hernández.

**Investigation:** Diana Riaño-Hernández, Angelli Ramírez-Conde, Paula A. Atehortua-Rivera.

**Methodology:** Diana Riaño-Hernández, Iwin Leenen.

**Project administration:** Diana Riaño-Hernández.

**Resources:** Diana Riaño-Hernández, Angelli Ramírez-Conde, Paula A. Atehortua-Rivera.

**Supervision:** Diana Riaño-Hernández.

**Validation:** Iwin Leenen.

**Writing – original draft:** Diana Riaño-Hernández, Iwin Leenen, Angelli Ramírez-Conde, Paula A. Atehortua-Rivera, José A. Piqueras.

**Writing – review & editing:** Iwin Leenen, José A. Piqueras.

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
