## [Decision Letter · Decision Letter 0]

16 Jul 2024

PONE-D-24-25194Validity evidence for assessing social-emotional psychological strengths in Colombian adolescents using the SEHS-SPLOS ONE

Dear Dr. Leenen,

Thank you for submitting your manuscript to PLOS ONE. After careful consideration, we feel that it has merit but does not fully meet PLOS ONE’s publication criteria as it currently stands. Therefore, we invite you to submit a revised version of the manuscript that addresses the points raised during the review process.

Please address reviewers’ comments. Please submit your revised manuscript by Aug 30 2024 11:59PM. If you will need more time than this to complete your revisions, please reply to this message or contact the journal office at plosone@plos.org. Please include the following items when submitting your revised manuscript:A rebuttal letter that responds to each point raised by the academic editor and reviewer(s). You should upload this letter as a separate file labeled 'Response to Reviewers'.A marked-up copy of your manuscript that highlights changes made to the original version. You should upload this as a separate file labeled 'Revised Manuscript with Track Changes'.An unmarked version of your revised paper without tracked changes. You should upload this as a separate file labeled 'Manuscript'.

We look forward to receiving your revised manuscript.

Kind regards,

Majed Sulaiman Alamri, PhD

Academic Editor

PLOS ONE

Journal Requirements:

4. You indicated that you had ethical approval for your study. In your Methods section, please ensure you have also stated whether you obtained consent from parents or guardians of the minors included in the study or whether the research ethics committee or IRB specifically waived the need for their consent.

6. Thank you for stating the following financial disclosure: 

"This research received funding from the Universidad Católica de Pereira and the Ministerio de Ciencia y Tecnología (DRH; Project 2019/850)."

Reviewers' comments:

Reviewer's Responses to Questions

**Comments to the Author**

1. Is the manuscript technically sound, and do the data support the conclusions?

Reviewer #1: Yes

Reviewer #2: Partly

2. Has the statistical analysis been performed appropriately and rigorously? 

Reviewer #1: Yes

Reviewer #2: No

3. Have the authors made all data underlying the findings in their manuscript fully available?

Reviewer #1: Yes

Reviewer #2: Yes

4. Is the manuscript presented in an intelligible fashion and written in standard English?

Reviewer #1: Yes

Reviewer #2: No

5. Review Comments to the Author

Reviewer #1: Thank you for giving me the opportunity to review your work titled "Validity evidence for assessing social-emotional psychological strengths in Colombian adolescents using the SEHS-S". It is a significant contribution in the field of psychology and the assessment of socio-emotional skills in adolescents.

Strengths

Validity and Robustness of the Model: The hierarchical model of the covitality construct is well-supported through confirmatory factor analyses.

Measurement Invariance: Metric and configural invariance have been confirmed across gender and age groups.

Cultural Application: The SEHS-S has been validated for the Colombian adolescent population, contributing to the validity evidence in different cultural contexts.

Significant Correlations: Moderate to high correlations with related constructs strengthen the convergent validity of the SEHS-S.

Large and Diverse Sample: The large sample of 1461 students provides a solid basis for statistical analyses.

Practical Implications: The article highlights the utility of the SEHS-S to identify specific needs and propose targeted intervention strategies to improve the socio-emotional well-being of adolescents.

Redundant Parts to Remove and Proposed Modifications

Abstract:

Lines to remove: 33-34, 35-36

Remove: "although differences in item functioning across gender and age were found. Furthermore, Covitality maintained moderate to high correlations with related constructs. As a conclusion, the SEHS-S can be considered a valid tool to assess psychological strengths, well-being, and resilience (i.e., Covitality) in Colombian adolescents. However, further research is needed to explore the differences in item functioning."

Replace with: "though further research is needed to explore item functioning differences across gender and age."

Introduction:

Lines to remove: 47-50

Remove: "Battling these adversities requires the development of strategies that identify emotional and behavioral issues in young individuals and provide support by strengthening their social-emotional growth at early stages and, particularly, with a focus on proactive prevention rather than relying on reactive approaches."

Replace with: "Developing strategies that identify and address emotional and behavioral issues early, with a focus on proactive prevention, is crucial."

Materials and Methods:

Lines to remove: 163-171

Remove: "First, we made the necessary translations and linguistic adjustments to Colombian Spanish for each of the above-mentioned instruments, based on expert judgments and cognitive interviews with Colombian adolescents. Subsequently, the project was presented in the participating schools and after informed consent was obtained from the adolescents and their parents, the four instruments, together with a 15-item sociodemographic questionnaire (asking about age, grade, and sex, amongst others), were applied in groups to the students that were present at school that day. Students were given a brief description of the project and an explanation of the instructions to fill out the questionnaires. The data collection took place in 2019 between March 1 and October 17."

Replace with: "Translations and linguistic adjustments to Colombian Spanish were made. After obtaining informed consent, the instruments and a sociodemographic questionnaire were administered in groups between March 1 and October 17, 2019."

Results:

Lines to remove: 271-273

Remove: "In interpreting the results in Table 4, recall that the SEHS‑S items are scored on a scale from 1 to 4 points and that the intercepts indicate the expected score for an individual with average scores (i.e., equal to 0) on the latent (first-order, second-order, and third-order) factors."

Replace with: "The SEHS‑S items are scored from 1 to 4 points, and the intercepts indicate the expected score for an individual with average latent factor scores."

We invite you to consult the article by Diotaiuti et al. (2017), "Alcohol drinking patterns in young people: A survey-based study" published in the Journal of Health Psychology, for a detailed analysis of youth behaviors that could enrich the discussion on the importance of psychological strengths and socio-emotional behaviors in adolescents. You can insert this citation in the Introduction section, in the paragraph discussing the importance of understanding the psychological and socio-emotional behaviors of adolescents.

Example of Insertion:

"Understanding the behavioral patterns and psychological strengths of adolescents is crucial for developing effective interventions. We invite you to consult the article by Diotaiuti et al. (2017) for a detailed analysis of youth behaviors that could enrich this discussion (Diotaiuti, 2017)."

These proposed changes aim to make the article more concise and clear, eliminating redundancies and improving overall readability. I hope these revisions are helpful and further enhance the quality of your work. If there are any further questions or requests for clarification, I am at your disposal.

Reviewer #2: ID: PONE-D-24-25194

Title:Validity evidence for assessing social-emotional psychological strengths in Colombian adolescents using the SEHS-S

Thank you for providing a chance to review this manuscript.

Detailed information:

Abstract

1) Abstract requires a brief overview of the background, purpose, methods, results, and conclusions of the article. It is recommended to add subheadings to make the structure of the article clearer.

2) The results section of the abstract is to be narrated with relevant statistical data, and it is recommended that the results section be rewritten with reference to high quality literature.

Materials and methods

Participants

Line104-105, page 5: “four public and rural secondary schools” How were these middle schools chosen? Why were these four middle schools chosen? Can you tell us the exact process.

Overall: 1) Is the 1461 sample size the final sample size? 2) 1461 Is the sample size sufficient? What formula was used to calculate the minimum sample size? 3) During the questionnaire collection process, were there any missing rates and what were they? What was done about the missing data. 4) Are there inclusion and exclusion criteria for the population? If so, please specify in the article.

Procedure

Line164-166, page 8: “necessary translations and linguistic adjustments to Colombian Spanish for each of the above-mentioned instruments” Can you be more specific about the process?

Line169-170, page 8: “were applied in groups to the students that were present at school that day” Is it all the students who were in school that day? Or was the sampling done through some sort of sampling method? Please elaborate.

Line171-172, page 8: “The data collection took place in 2019 between March 1 and Octobre 17. ” Can you state the collection time for each school separately?

Discussion

1) What are the limitations of this paper?

Conclusions

1) It is suggested that the conclusions be shortened.

This paper presents a comprehensive overview of the validity evidence obtained through the SEHS-S for the Common Vitality Assessment for Colombian adolescents, which has some theoretical value and practical implications. However, there are some problems with this paper that require revision by the authors.

Thank you and my best,

Your reviewer

6. PLOS authors have the option to publish the peer review history of their article (what does this mean?). If published, this will include your full peer review and any attached files.

Reviewer #1: **Yes: **I want mine to be a public review. insert my name Pierluigi Diotaiuti

Reviewer #2: No

---

## [Author Response · Author response to Decision Letter 0]

23 Aug 2024

I have uploaded files "Additional Journal Requirements" and "Response te Reviewers" that provide the details of how we responded to the specific comments by the editor and the reviewers. I am unsure whether I have to copy these responses in this box or whether it is sufficient to refer to these documents.

---

## [Decision Letter · Decision Letter 1]

8 Sep 2024

PONE-D-24-25194R1Validity evidence for assessing social-emotional psychological strengths in Colombian adolescents using the SEHS-SPLOS ONE

Dear Dr. Leenen,

Thank you for submitting your manuscript to PLOS ONE. After careful consideration, we feel that it has merit but does not fully meet PLOS ONE’s publication criteria as it currently stands. Therefore, we invite you to submit a revised version of the manuscript that addresses the points raised during the review process.

Please address reviewers’ comments. Please submit your revised manuscript by Oct 23 2024 11:59PM. If you will need more time than this to complete your revisions, please reply to this message or contact the journal office at plosone@plos.org. Please include the following items when submitting your revised manuscript:A rebuttal letter that responds to each point raised by the academic editor and reviewer(s). You should upload this letter as a separate file labeled 'Response to Reviewers'.A marked-up copy of your manuscript that highlights changes made to the original version. You should upload this as a separate file labeled 'Revised Manuscript with Track Changes'.An unmarked version of your revised paper without tracked changes. You should upload this as a separate file labeled 'Manuscript'.If applicable, we recommend that you deposit your laboratory protocols in protocols.io to enhance the reproducibility of your results. Protocols.io assigns your protocol its own identifier (DOI) so that it can be cited independently in the future. For instructions see: https://journals.plos.org/plosone/s/submission-guidelines#loc-laboratory-protocols. Additionally, PLOS ONE offers an option for publishing peer-reviewed Lab Protocol articles, which describe protocols hosted on protocols.io. Read more information on sharing protocols at https://plos.org/protocols?utm_medium=editorial-email&utm_source=authorletters&utm_campaign=protocols.

We look forward to receiving your revised manuscript.

Kind regards,

Majed Sulaiman Alamri, PhD

Academic Editor

PLOS ONE

Journal Requirements:

Reviewers' comments:

Reviewer's Responses to Questions

**Comments to the Author**

1. If the authors have adequately addressed your comments raised in a previous round of review and you feel that this manuscript is now acceptable for publication, you may indicate that here to bypass the “Comments to the Author” section, enter your conflict of interest statement in the “Confidential to Editor” section, and submit your "Accept" recommendation.

Reviewer #1: All comments have been addressed

Reviewer #2: (No Response)

2. Is the manuscript technically sound, and do the data support the conclusions?

Reviewer #1: Yes

Reviewer #2: Partly

3. Has the statistical analysis been performed appropriately and rigorously? 

Reviewer #1: Yes

Reviewer #2: N/A

4. Have the authors made all data underlying the findings in their manuscript fully available?

Reviewer #1: Yes

Reviewer #2: Yes

5. Is the manuscript presented in an intelligible fashion and written in standard English?

Reviewer #1: Yes

Reviewer #2: Yes

6. Review Comments to the Author

Reviewer #1: Thank you very much for the opportunity to review this well-conducted study. I particularly appreciated the innovative approach and the detailed attention you demonstrated in assessing the socio-emotional psychological strengths of Colombian adolescents through the Covitality model. It is clear that you have put great effort into producing a rigorous and culturally meaningful research. My heartfelt congratulations to all the authors for the excellent quality of the work.

The manuscript presents several noteworthy aspects that deserve to be highlighted:

Solid theoretical approach: The description of the Covitality model and its application in different contexts is clearly presented, highlighting the broad scope of the construct and its relevance in the field of mental health.

Rigorous statistical methodology: The use of confirmatory factor analyses (CFA) and the verification of gender and age invariance are effective tools for assessing the validity of the scale used.

Contribution to local literature: The validation of SEHS-S in the Colombian context is a valuable contribution to positive psychology in Latin America, where similar instruments are scarce.

Balanced discussion: The reflection on the results, particularly on gender and age differences, is well-founded and provides an important basis for future targeted interventions.

Suggested Revisions

Abstract

Line 35: Add numerical values for the correlations to improve clarity. Example: "Covitality maintained moderate (r = 0.5) to high (r = 0.7) correlations with related constructs."

Introduction

Line 47: I suggest expanding the description of prevention strategies with concrete examples of interventions based on socio-emotional strengths, such as school or community programs.

Line 61: The phrase "psychological anxiety" could be more specific, clarifying whether it refers to specific anxiety disorders or a more general condition.

Materials and Methods

Line 104: Add a brief note on the criteria for selecting the participating schools to improve the transparency of the sampling. Example: "Schools were selected based on their geographic and socioeconomic representativeness within the department of Risaralda."

Line 172: Correct "Octobre" to "October."

Line 183: Providing a brief justification for using SAS V9.4 could help clarify the software choice. Example: "SAS V9.4 was chosen for its robustness in multivariate data analysis and its ability to handle large datasets with missing values."

Results

Line 226: Quantify the statement "relatively high nonresponse rate" for greater clarity. Example: "The nonresponse rate was 22.7% for the TEIQue-ASF, the highest among the instruments used."

Lines 225-227: Briefly discuss the potential impact of the nonresponse rate on the study's conclusions. Example: "Although the overall nonresponse rate was low, its concentration in longer instruments may have influenced the results, particularly for the SEHS-S."

Discussion

Line 333: Expand the comparison with similar studies in other cultural contexts. Example: "These findings are consistent with those obtained in studies conducted in Spain and Japan, where the hierarchical structure of the Covitality model has been confirmed, albeit with slight variations in the lower-order factors."

Line 370: Replace "further investigated in future research" with a more specific phrase. Example: "Further investigations, particularly longitudinal studies, will be needed to explore how gender and age differences evolve over time."

Despite the considerable value of this work, there are some limitations that should be addressed to ensure greater transparency and completeness of the manuscript:

Self-report bias: Given the self-report nature of the instruments used (SEHS-S and other questionnaires), there is a potential risk of bias related to social desirability or the adolescents’ ability to accurately assess their own emotional state. This aspect should be acknowledged and discussed, as it may affect the results, especially in the measurement of subjective well-being.

Limited sample representativeness: Although the sample is large, it focuses mainly on adolescents from rural and socioeconomically disadvantaged backgrounds in a specific region of Colombia. This may limit the generalizability of the results to other geographical areas or adolescents from different socioeconomic groups. Mentioning this limitation would help clarify the boundaries of the study’s applicability.

Lack of longitudinal data: The study provides a cross-sectional snapshot of SEHS-S validity. However, it would be interesting to explore how socio-emotional psychological strengths develop over time in adolescents. Longitudinal studies could clarify the evolution of the gender and age differences found.

Unresolved scalar invariance: The lack of scalar invariance in some SEHS-S items between gender and age groups limits the ability to make direct comparisons between these groups. Although this has been discussed, further emphasis on how this impacts data interpretation would be beneficial.

Sections to Eliminate

There are no sections that need complete elimination, but some parts could be streamlined for greater clarity. For example:

The sections between lines 330-350 on the confirmation of the factor model could be condensed to avoid repetition while keeping the essential points.

I suggest that the authors cite "A Structural Model of Self-Efficacy in Handball Referees" by Pierluigi Diotaiuti and colleagues, published in Frontiers in Psychology (2017). This article focuses on the role of self-efficacy in sports contexts and could be relevant to your study on psychological strengths in adolescents, particularly when discussing the importance of self-belief and emotional competence as factors contributing to resilience and well-being. The framework of self-efficacy in the referenced article aligns with themes of personal psychological strengths that could enrich your discussion.

In the Introduction, after discussing the Covitality model and its components (lines 56-58), you could introduce this citation when expanding on the significance of self-efficacy as a psychological strength that promotes resilience in various contexts.

Example of Citation:

You could say: "The importance of self-efficacy in fostering psychological resilience has been demonstrated across different populations, including in sports contexts, where it has been shown to significantly impact performance and coping mechanisms (Diotaiuti et al., 2017)."

This will strengthen the theoretical framework of your study by linking it to relevant research on psychological strengths in non-clinical settings.

Reference: Diotaiuti, P., Falese, L., Mancone, S., & Purromuto, F. (2017). A structural Model of Self-efficacy in Handball Referees. Frontiers in psychology, 8, 811. https://doi.org/10.3389/fpsyg.2017.00811

Reviewer #2: ID: PONE-D-24-25194R1

Title: Validity evidence for assessing social-emotional psychological strengths in Colombian adolescents using the SEHS-S

Thank you for providing a chance to review this manuscript.

Recommendation: Minor.

Materials and methods

1) “The inclusion criteria, which were met by all participants, required a minimal level of cognitive ability and reading skills to complete the self-report questionnaires. ” What were the inclusion and exclusion criteria for participants and please specify. How can it be demonstrated that the participant has a minimum level of cognitive and reading ability to complete the questionnaire?

2) What is the total sample size for the final fill? How many of these were valid? Were there any invalid questionnaires in this survey? How were invalid questionnaires handled?

As a result of the author's revisions, the article has become progressively standardized and acceptable, however, there are still some minor problems, so the author is requested to revise it carefully.

Thank you and my best,

Your reviewer

7. PLOS authors have the option to publish the peer review history of their article (what does this mean?). If published, this will include your full peer review and any attached files.

Reviewer #1: **Yes: **Pierluigi Diotaiuti

Reviewer #2: No

---

## [Author Response · Author response to Decision Letter 1]

30 Oct 2024

Please review the document "Response to Reviewers".

---

## [Decision Letter · Decision Letter 2]

12 Nov 2024

Validity evidence for assessing social-emotional psychological strengths in Colombian adolescents using the SEHS-S

PONE-D-24-25194R2

Dear Dr. Leenen,

We’re pleased to inform you that your manuscript has been judged scientifically suitable for publication and will be formally accepted for publication once it meets all outstanding technical requirements.

Kind regards,

Majed Sulaiman Alamri, PhD

Academic Editor

PLOS ONE

Additional Editor Comments (optional):

Reviewers' comments:

Reviewer's Responses to Questions

**Comments to the Author**

1. If the authors have adequately addressed your comments raised in a previous round of review and you feel that this manuscript is now acceptable for publication, you may indicate that here to bypass the “Comments to the Author” section, enter your conflict of interest statement in the “Confidential to Editor” section, and submit your "Accept" recommendation.

Reviewer #1: All comments have been addressed

Reviewer #2: All comments have been addressed

2. Is the manuscript technically sound, and do the data support the conclusions?

Reviewer #1: Yes

Reviewer #2: Partly

3. Has the statistical analysis been performed appropriately and rigorously? 

Reviewer #1: Yes

Reviewer #2: Yes

4. Have the authors made all data underlying the findings in their manuscript fully available?

Reviewer #1: Yes

Reviewer #2: Yes

5. Is the manuscript presented in an intelligible fashion and written in standard English?

Reviewer #1: Yes

Reviewer #2: Yes

6. Review Comments to the Author

Reviewer #1: Dear Editor and Authors,

After a careful review of the manuscript and the modifications the authors have implemented following our suggestions, we are pleased to inform you that the article has now reached a level of quality and completeness that meets the requirements for publication.

The proposed revisions have been effectively incorporated, enhancing the clarity, coherence, and depth of the content. The structure of the article is now solid and well-organized, with arguments thoroughly developed and supported by appropriate evidence. Additionally, the authors have diligently addressed the points raised during the review process, demonstrating particular attention to scientific accuracy and clarity of presentation.

Given the results achieved and the high quality of the content, we believe the article is now ready for publication. We are confident that this work will make a significant contribution to the literature in the field and will be of great interest to the scientific community.

Reviewer #2: ID: PONE-D-24-25194R2

Title: Validity evidence for assessing social-emotional psychological strengths in Colombianadolescents using the SEHS-S

Thank you for providing a chance to review this manuscript.

Recommendation: Accept.

Detailed information:

After careful revision by the authors, the manuscript has met the basic requirements for publication, congratulations!

Thank you and my best,

Your reviewer

7. PLOS authors have the option to publish the peer review history of their article (what does this mean?). If published, this will include your full peer review and any attached files.

Reviewer #1: **Yes: **Pierluigi Diotaiuti

Reviewer #2: No

---

## [Editor Report · Acceptance letter]

18 Nov 2024

PONE-D-24-25194R2 

PLOS ONE

Dear Dr. Leenen, 

I'm pleased to inform you that your manuscript has been deemed suitable for publication in PLOS ONE. Congratulations! Your manuscript is now being handed over to our production team.

Kind regards, 

on behalf of

Dr. Majed Sulaiman Alamri 

Academic Editor

PLOS ONE